# Optimal Variance Control of the Score Function Gradient Estimator for Importance Weighted Bounds

**Valentin Liévin** [1]   **Andrea Dittadi**[1]   **Anders Christensen**[1]   **Ole Winther**[1, 2, 3]

[1] Section for Cognitive Systems, Technical University of Denmark
[2] Bioinformatics Centre, Department of Biology, University of Copenhagen
[3] Centre for Genomic Medicine, Rigshospitalet, Copenhagen University Hospital
{valv,adit}@dtu.dk, anders.christensen321@gmail.com, olwi@dtu.dk

## Abstract

This paper introduces novel results for the score function gradient estimator of the importance weighted variational bound (IWAE). We prove that in the limit of large $K$ (number of importance samples) one can choose the control variate such that the Signal-to-Noise ratio (SNR) of the estimator grows as $\sqrt{K}$. This is in contrast to the standard pathwise gradient estimator where the SNR decreases as $1/\sqrt{K}$. Based on our theoretical findings we develop a novel control variate that extends on VIMCO. Empirically, for the training of both continuous and discrete generative models, the proposed method yields superior variance reduction, resulting in an SNR for IWAE that increases with $K$ without relying on the reparameterization trick. The novel estimator is competitive with state-of-the-art reparameterization-free gradient estimators such as Reweighted Wake-Sleep (RWS) and the thermodynamic variational objective (TVO) when training generative models.

## 1 Introduction

Gradient-based learning is now widespread in the field of machine learning, in which recent advances have mostly relied on the backpropagation algorithm, the workhorse of modern deep learning. In many instances, for example in the context of unsupervised learning, it is desirable to make models more expressive by introducing stochastic latent variables. Backpropagation thus has to be augmented with methodologies for marginalization over latent variables.

Variational inference using an inference model (amortized inference) has emerged as a key method for training and inference in latent variable models [1–7]. The pathwise gradient estimator, based on the reparameterization trick [2, 3], often gives low-variance estimates of the gradient for continuous distributions. However, since discrete distributions cannot be reparameterized, these methods are not applicable to inference in complex simulators with discrete variables, such as reinforcement learning or advanced generative processes [8–11]. While the score function (or Reinforce) estimator [12] is more generally applicable, it is well known to suffer from large variance. Consequently, most of the recent developments focus on reducing the variance using control variates [13–18] and using alternative variational objectives [9, 19–21].

Recently, variational objectives tighter than the traditional evidence lower bound (ELBO) have been proposed [21, 22]. In importance weighted autoencoders (IWAE) [22] the tighter bound comes with the price of a $K$-fold increase in the required number of samples from the inference network. Despite yielding a tighter bound, using more samples can be detrimental to the learning of the inference model [23]. In fact, the Signal-to-Noise ratio (the ratio of the expected gradient to its standard deviation) of the pathwise estimator has been shown to decrease at a rate $\mathcal{O}(K^{-1/2})$ [23]. Although this can be improved to $\mathcal{O}(K^{1/2})$ by exploiting properties of the gradient to cancel high-variance

terms [24], the variational distributions are still required to be reparameterizable. In this work we introduce OVIS (*Optimal Variance – Importance Sampling*), a novel score function-based estimator for importance weighted objectives with improved SNR.

The main contributions of this paper are: 1) A proof that, with an appropriate choice of control variate, the score function estimator for the IWAE objective can achieve a Signal-to-Noise Ratio $\text{SNR} = \mathcal{O}(K^{1/2})$ as the number of importance samples $K \to \infty$. 2) A derivation of OVIS, a class of practical low-variance score function estimators following the principles of our theoretical analysis. 3) State-of-the-art results on a number of non-trivial benchmarks for both discrete and continuous stochastic variables, with comparison to a range of recently proposed score function methods.

## 2  Optimizing the Importance Weighted Bound

**Importance weighted bound (IWAE)**   Amortized variational inference allows fitting a latent variable model $p_\theta(\mathbf{x}, \mathbf{z})$ to the data using an approximate posterior $q_\phi(\mathbf{z}|\mathbf{x})$ [2]. By using multiple importance weighted samples, we can derive a lower bound to the log marginal likelihood that is uniformly tighter as the number of samples, $K$, increases [22]. The *importance weighted bound* (IWAE) for one data point $\mathbf{x}$ is:

$$\mathcal{L}_K(\mathbf{x}) := \mathbb{E}\left[\log \hat{Z}\right] \qquad \hat{Z} := \frac{1}{K}\sum_{k=1}^{K} w_k \qquad w_k := \frac{p_\theta(\mathbf{x}, \mathbf{z}_k)}{q_\phi(\mathbf{z}_k|\mathbf{x})} \,, \tag{1}$$

where $\mathbb{E}$ denotes an expectation over the $K$-copy variational posterior $q_\phi(\mathbf{z}_{1:K}|\mathbf{x}) := \prod_{k=1}^{K} q_\phi(\mathbf{z}_k|\mathbf{x})$. This bound coincides with the traditional evidence lower bound (ELBO) for $K = 1$. The log likelihood lower bound for the entire data set is $\mathcal{L}_K(\mathbf{x}_{1:n}) = \sum_{i=1}^{n} \mathcal{L}_K(\mathbf{x}_i)$. In the following we will derive results for one term $\mathcal{L}_K = \mathcal{L}_K(\mathbf{x})$.

**Score function estimator**   Without making assumptions about the variational distribution, the gradient of the importance weighted bound (1) with respect to the parameters of the approximate posterior factorizes as (see Appendix A):

$$\nabla_\phi \mathcal{L}_K = \mathbb{E}\left[\sum_k d_k \mathbf{h}_k\right] \qquad d_k := \log \hat{Z} - v_k \qquad v_k := \frac{w_k}{\sum_{l=1}^{K} w_l} \,, \tag{2}$$

where $\mathbf{h}_k := \nabla_\phi \log q_\phi(\mathbf{z}_k|\mathbf{x})$ is the score function. A Monte Carlo estimate of the expectation in (2) yields the *score function* (or *Reinforce*) *estimator*.

**Control variates**   The vanilla score function estimator of (2) is often not useful in practice due to its large sample-to-sample variance. By introducing control variates that aim to cancel out zero expectation terms, this variance can be reduced while keeping the estimator unbiased.

Given posterior samples $\mathbf{z}_1, \ldots, \mathbf{z}_K \sim q_\phi(\mathbf{z}_{1:K}|\mathbf{x})$, let $\mathbf{z}_{-k}$ denote $[\mathbf{z}_1, \ldots, \mathbf{z}_{k-1}, \mathbf{z}_{k+1}, \ldots, \mathbf{z}_K]$, let $\mathbb{E}_k[\ldots]$ and $\mathbb{E}_{-k}[\ldots]$ be the expectations over the variational distributions of $\mathbf{z}_k$ and $\mathbf{z}_{-k}$, respectively, and let $\{c_k\}_{k=1}^{K}$ be scalar control variates, with each $c_k = c_k(\mathbf{z}_{-k})$ independent of $\mathbf{z}_k$. Using the independence of $c_k$ and $\mathbf{h}_k$ for each $k$, and the fact that the score function has zero expectation, we have $\mathbb{E}[c_k \mathbf{h}_k] = \mathbb{E}_{-k}[c_k]\mathbb{E}_k[\mathbf{h}_k] = 0$. Thus, we can define an unbiased estimator of (2) as:

$$\mathbf{g} := \sum_k (d_k - c_k)\,\mathbf{h}_k \tag{3}$$

$$\mathbb{E}[\mathbf{g}] = \mathbb{E}\left[\sum_k (d_k - c_k)\,\mathbf{h}_k\right] = \mathbb{E}\left[\sum_k d_k \mathbf{h}_k\right] = \nabla_\phi \mathcal{L}_K \,. \tag{4}$$

In the remainder of this paper, we will use the decomposition $d_k = f_k + f_{-k}$, where $f_k = f_k(\mathbf{z}_k, \mathbf{z}_{-k})$ and $f_{-k} = f_{-k}(\mathbf{z}_{-k})$ denote terms that depend and do not depend on $\mathbf{z}_k$, respectively. This will allow us to exploit the mutual independence of $\{\mathbf{z}_k\}_{k=1}^{K}$ to derive optimal control variates.

**Signal-to-Noise Ratio (SNR)**   We will compare the different estimators on the basis of their Signal-to-noise ratio. Following [23], we define the SNR for each component of the gradient vector as

$$\text{SNR}_i := \frac{|\mathbb{E}[g_i]|}{\sqrt{\text{Var}[g_i]}} \,, \tag{5}$$

where $g_i$ denotes the $i$th component of the gradient vector.

In Section 3 we derive the theoretical SNR for the optimal choice of control variates in the limit $K \to \infty$. In Section 4 we derive the optimal scalar control variates $\{c_k\}_{k=1}^K$ by optimizing the trace of the covariance of the gradient estimator $\mathbf{g}$, and in Section 6 we experimentally compare our approach with state-of-the-art gradient estimators in terms of SNR.

## 3 Asymptotic Analysis of the Signal-to-Noise Ratio

Assuming the importance weights have finite variance, i.e. $\mathrm{Var}[w_k] < \infty$, we can derive the asymptotic behavior of the SNR as $K \to \infty$ by expanding $\log \hat{Z}$ as a Taylor series around $Z := p_\theta(\mathbf{x}) = \int p_\theta(\mathbf{x}, \mathbf{z}) d\mathbf{z}$ [23]. A direct application of the pathwise gradient estimator (reparameterization trick) to the importance weighted bound results in an SNR that scales as $\mathcal{O}(K^{-1/2})$ [23], which can be improved to $\mathcal{O}(K^{1/2})$ by exploiting properties of the gradient [24]. In the following we will show that, for a specific choice of control variate, the SNR of the score function estimator scales as $\mathcal{O}(K^{1/2})$. Thus, a score function estimator exists for which *increasing the number of importance samples benefits the gradient estimate of the parameters of the variational distribution.*

For the asymptotic analysis we rewrite the estimator as $\mathbf{g} = \sum_k \left( -\frac{\partial \log \hat{Z}}{\partial w_k} w_k + \log \hat{Z} - c_k \right) \mathbf{h}_k$ and apply a second-order Taylor expansion to $\log \hat{Z}$. The resulting expression $\mathbf{g} = \sum_k (f_k + f_{-k} - c_k) \mathbf{h}_k$ separates terms $f_k$ that contribute to the expected gradient from terms $f_{-k}$ that have zero expectation and thus only contribute to the variance (cf. Appendix B):

$$f_k \approx \frac{w_k^2}{2K^2 Z^2} \tag{6}$$

$$f_{-k} \approx \log Z - \frac{3}{2} + \frac{2}{KZ} \sum_{l \neq k} w_l - \frac{1}{2K^2 Z^2} \left( \sum_{l \neq k} w_l \right)^2 . \tag{7}$$

Since $f_{-k}$ and $c_k$ are independent of $\mathbf{h}_k$, the expected gradient is (cf. Appendix C.1):

$$\mathbb{E}[\mathbf{g}] = \sum_k \mathbb{E}[f_k \mathbf{h}_k] \approx \frac{1}{2Z^2 K} \mathbb{E}_1 \left[ w_1^2 \mathbf{h}_1 \right] = \mathcal{O}(K^{-1}) , \tag{8}$$

where $\mathbb{E}_1$ denotes an expectation over the first latent distribution $q_\phi(\mathbf{z}_1 | \mathbf{x})$. Since the choice of control variates $c_k = c_k(\mathbf{z}_{-k})$ is free, we can choose $c_k = f_{-k}$ to cancel out all zero expectation terms. The resulting covariance, derived in Appendix C.2, is:

$$\mathrm{Cov}[\mathbf{g}] = \mathrm{Cov} \left[ \sum_k f_k \mathbf{h}_k \right] \approx \frac{1}{4K^3 Z^4} \mathrm{Cov}_1 \left[ w_1^2 \mathbf{h}_1 \right] = \mathcal{O}(K^{-3}) \tag{9}$$

with $\mathrm{Cov}_1$ indicating the covariance over $q_\phi(\mathbf{z}_1 | \mathbf{x})$. Although as we discuss in Section 4 this is not the minimal variance choice of control variates, it is sufficient to achieve an SNR of $\mathcal{O}(K^{1/2})$.

## 4 Optimal Control Variate

The analysis above shows that in theory it is possible to attain a good SNR with the score function estimator. In this section we derive the optimal (in terms of variance of the resulting estimator) control variates $\{c_k\}_{k=1}^K$ by decomposing $\mathbf{g} = \sum_k (f_k + f_{-k} - c_k) \mathbf{h}_k$ as above, and minimizing the trace of the covariance matrix, i.e. $\mathbb{E}[||\mathbf{g}||^2] - ||\mathbb{E}[\mathbf{g}]||^2$. Since $\mathbb{E}[f_{-k} \mathbf{h}_k]$ and $\mathbb{E}[c_k \mathbf{h}_k]$ are both zero, $\mathbb{E}[\mathbf{g}] = \nabla_\phi \mathcal{L}_K$ does not depend on $c_k$. Thus, the minimization only involves the first term:

$$\frac{1}{2} \frac{\partial}{\partial c_k} \mathbb{E} \left[ ||\mathbf{g}||^2 \right] = \mathbb{E} \left[ \mathbf{h}_k^T \sum_l (f_l + f_{-l} - c_l) \mathbf{h}_l \right]$$

$$= \mathbb{E}_{-k} \left[ \sum_l \mathbb{E}_k \left[ f_l \mathbf{h}_k^T \mathbf{h}_l \right] + (f_{-k} - c_k) \mathbb{E}_k \left[ ||\mathbf{h}_k||^2 \right] \right] .$$

where $\mathbb{E}_k$ and $\mathbb{E}_{-k}$ indicate expectations over $q_\phi(\mathbf{z}_k | \mathbf{x})$ and $q_\phi(\mathbf{z}_{-k} | \mathbf{x})$, respectively. Setting the argument of $\mathbb{E}_{-k}$ to zero, we get the optimal control variates $c_k = c_k(\mathbf{z}_{-k})$ and gradient estimator $\mathbf{g}$:

$$c_k = f_{-k} + \sum_l \frac{\mathbb{E}_k \left[ f_l \mathbf{h}_k^T \mathbf{h}_l \right]}{\mathbb{E}_k \left[ ||\mathbf{h}_k||^2 \right]} \tag{10}$$

$$\mathbf{g} = \sum_k \left( f_k - \sum_l \frac{\mathbb{E}_k \left[ f_l \mathbf{h}_k^T \mathbf{h}_l \right]}{\mathbb{E}_k \left[ ||\mathbf{h}_k||^2 \right]} \right) \mathbf{h}_k . \tag{11}$$

Applying (11) in practice requires marginalizing over one latent variable and decoupling terms that do not depend on $\mathbf{z}_k$ from those that do. In the remainder of this section we will 1) make a series of approximations to keep computation tractable, and 2) consider two limiting cases for the *effective sample size* (ESS) [25] in which we can decouple terms.

**Simplifying approximations to Equation (11)** First, we consider a term with $l \neq k$, define $\Delta f_l := f_l - \mathbb{E}_k[f_l]$, and subtract and add $\mathbb{E}_k[f_l]$ from inside the expectation:

$$\mathbb{E}_k\left[f_l\mathbf{h}_k^T\right]\mathbf{h}_l = \mathbb{E}_k\left[\Delta f_l\mathbf{h}_k^T\right]\mathbf{h}_l + \mathbb{E}_k[f_l]\mathbb{E}_k\left[\mathbf{h}_k^T\right]\mathbf{h}_l = \mathbb{E}_k\left[\Delta f_l\mathbf{h}_k^T\right]\mathbf{h}_l$$

where we used the fact that $\mathbb{E}_k\left[\mathbf{h}_k\right] = 0$. The $l \neq k$ terms thus only contribute to fluctuations relative to a mean value, and we assume they can be neglected.

Second, we assume that $|\phi|$, the number of parameters of $q_\phi$, is large, and the terms of the sum $\|\mathbf{h}_k\|^2 = \sum_{i=1}^{|\phi|} h_{ki}^2$ are approximately independent with finite variances $\sigma_i^2$. By the Central Limit Theorem we approximate the distribution of $\Delta\|\mathbf{h}_k\|^2 := \|\mathbf{h}_k\|^2 - \mathbb{E}_k\left[\|\mathbf{h}_k\|^2\right]$ with a zero-mean Gaussian with standard deviation $\left(\sum_{i=1}^{|\phi|}\sigma_i^2\right)^{1/2}$. Seeing that $\mathbb{E}_k\left[\|\mathbf{h}_k\|^2\right]$ is $\mathcal{O}(|\phi|)$, we have

$$\frac{\mathbb{E}_k\left[f_k\|\mathbf{h}_k\|^2\right]}{\mathbb{E}_k\left[\|\mathbf{h}_k\|^2\right]} = \mathbb{E}_k\left[f_k\right] + \frac{\mathbb{E}_k\left[f_k\Delta\|\mathbf{h}_k\|^2\right]}{\mathbb{E}_k\left[\|\mathbf{h}_k\|^2\right]} = \mathbb{E}_k\left[f_k\right] + \mathcal{O}(|\phi|^{-1/2}),$$

where we used that the argument in the numerator scales as $\left(\sum_{i=1}^{|\phi|}\sigma_i^2\right)^{1/2} = \mathcal{O}(|\phi|^{1/2})$.

Finally, the expectation can be approximated with a sample average. Writing $f_k = f_k(\mathbf{z}_k, \mathbf{z}_{-k})$ and drawing $S$ new samples $\mathbf{z}^{(1)}, \dots, \mathbf{z}^{(S)} \sim q_\phi(\mathbf{z}|\mathbf{x})$:

$$\mathbb{E}_k\left[f_k\right] \approx \frac{1}{S}\sum_{s=1}^{S}f_k(\mathbf{z}^{(s)}, \mathbf{z}_{-k}).$$

This will introduce additional fluctuations with scale $S^{-1/2}$.

Putting these three approximations together and using $d_k(\mathbf{z}_k, \mathbf{z}_{-k}) = f_k(\mathbf{z}_k, \mathbf{z}_{-k}) + f_{-k}(\mathbf{z}_{-k})$, we obtain the sample-based expression of the OVIS estimator, called $\text{OVIS}_{\text{MC}}$ in the following:

$$\text{OVIS}_{\text{MC}}: \quad \mathbf{g} \approx \sum_k \left(d_k(\mathbf{z}_k, \mathbf{z}_{-k}) - \frac{1}{S}\sum_{s=1}^{S}d_k(\mathbf{z}^{(s)}, \mathbf{z}_{-k})\right)\mathbf{h}_k. \tag{12}$$

Naively, this will produce a large computational overhead because we now have in total $KS$ terms. However, we can reduce this to $\mathcal{O}(K+S)$ because the bulk of the computation comes from evaluating the importance weights and because the $S$ auxiliary samples can be reused for all $K$ terms.

**Effective sample size (ESS)** The ESS [25] is a commonly used yardstick of the efficiency of an importance sampling estimate, defined as

$$\text{ESS} := \frac{\left(\sum_k w_k\right)^2}{\sum_k w_k^2} = \frac{1}{\sum_k v_k^2} \in [1, K]. \tag{13}$$

A low ESS occurs when only a few weights dominate, which indicates that the proposal distribution $q$ poorly matches $p$. In the opposite limit, the variance of importance weights is finite and the ESS will scale with $K$. Therefore the limit $\text{ESS} \gg 1$ corresponds to the asymptotic limit studied in Section 3.

**Optimal control for ESS limits and unified interpolation** In the following, we consider the two extreme limits $\text{ESS} \gg 1$ and $\text{ESS} \approx 1$ to derive sample-free approximations to the optimal control. We can thus in these limits avoid the sample fluctuations and excess computation of $\text{OVIS}_{\text{MC}}$.

We first consider $\text{ESS} \gg 1$ and for each $k$ we introduce the unnormalized leave-$w_k$-out approximation to $\hat{Z}$:

$$\widetilde{Z}_{[-k]} := \frac{1}{K}\sum_{l \neq k}w_l \quad \text{such that} \quad \hat{Z} - \widetilde{Z}_{[-k]} = \frac{w_k}{K}. \tag{14}$$

Assuming $\text{Var}[w_k] < \infty$, this difference is $\mathcal{O}(K^{-1})$ as $K \to \infty$, thus we can expand $\log \hat{Z}$ around $\hat{Z} = \widetilde{Z}_{[-k]}$. In this limit, the optimal control variate simplifies to (cf. Appendix D.1):

$$\text{ESS} \gg 1: \quad c_k \approx \log \frac{1}{K-1} \sum_{l \neq k} w_l + \log(1 - \frac{1}{K}) \,. \tag{15}$$

When $\text{ESS} \approx 1$, one weight is much larger than the others and the assumption above is no longer valid. To analyze this frequently occurring scenario, assume that $k' = \text{argmax}_l w_l$ and $w_{k'} \gg \sum_{l \neq k'} w_l$. In this limit $\log \hat{Z} \approx \log w_{k'}/K$ and $v_k \approx \delta_{k,k'}$ and thus $d_k = \log w_{k'}/K - \delta_{k,k'}$. In Appendix D.2 we show we can approximate Equation (10) with

$$\text{ESS} \approx 1: \quad c_k \approx \log \frac{1}{K-1} \sum_{l \neq k} w_l - v_k \,. \tag{16}$$

We introduce OVIS$_\sim$ to interpolate between the two limits (Appendix D.3):

$$c_k^\gamma := \log \frac{1}{K-1} \sum_{l \neq k} w_l - \gamma v_k + (1 - \gamma) \log \left(1 - \frac{1}{K}\right) \qquad \gamma \in [0, 1] \,. \tag{17}$$

In this paper we will only conduct experiments for the two limiting cases $\gamma = 0$, corresponding to Equation (15), and $\gamma = 1$ approximating Equation (16). Tuning the parameter $\gamma$ in the range $[0, 1]$ will be left for future work. We discuss the implementation in the appendix K.

**Higher ESS with looser lower bound** Empirically we observe that training may be impaired by a low ESS and by *posterior collapse* [4, 26–29]. This motivates trading the tight IWAE objective for a gradient estimator with higher ESS. To that end, we use the importance weighted Rényi (IWR) bound:

$$\mathcal{L}_K^\alpha(\mathbf{x}) := \frac{1}{1-\alpha} \mathbb{E}\left[\log \hat{Z}(\alpha)\right] \quad \hat{Z}(\alpha) := \frac{1}{K} \sum_k w_k^{1-\alpha} \tag{18}$$

which for $\alpha \in [0, 1]$ is a lower bound on the Rényi objective $\log \mathbb{E}_1\left[w_1^{1-\alpha}\right]/(1-\alpha)$ [30]. The Rényi objective in itself coincides with $\log p(\mathbf{x})$ for $\alpha = 0$ and is monotonically non-increasing in $\alpha$, i.e. is an evidence lower bound [30]. So we have a looser bound but higher $\text{ESS}(\alpha) = 1/\sum_k v_k^2(\alpha) \geq \text{ESS}(0)$ for $\alpha \in [0, 1]$ with $v_k(\alpha) = w_k^{1-\alpha}/\sum_l w_l^{1-\alpha}$. Furthermore, for $\alpha = 1$ the bound corresponds to the ELBO and the divergence $\mathcal{D}_{\text{KL}}(q_\phi(\mathbf{z}|\mathbf{x})||p_\theta(\mathbf{z}|\mathbf{x}))$ is guaranteed to be minimized. In Appendix E we derive the score function estimator and control variate expressions for $\mathcal{L}_K^\alpha$. The objective can either be used in a warm-up scheme by gradually decreasing $\alpha \to 0$ throughout iterations or can be run with a constant $0 < \alpha < 1$.

## 5 Related Work

The score function estimator with control variates can be used with all the commonly used variational families. By contrast, the reparameterization trick is only applicable under specific conditions. We now give a brief overview of the existing alternatives and refer the reader to [31] for a more extensive review. The importance of handling discrete distributions without relaxations is discussed in [9].

NVIL [13], DARN [17], and MuProp [18] demonstrate that score function estimators with carefully crafted control variates allow to train deep generative models. VIMCO [14] extends this to multi-sample objectives, and recycles the Monte Carlo samples $\mathbf{z}_{-k}$ to define a control variate $c_k = c_k(\mathbf{z}_{-k})$. Unlike OVIS, VIMCO only controls the variance of the term $\log \hat{Z}$ in $d_k = \log \hat{Z} - v_k$, leaving $v_k$ uncontrolled, and causing the SNR to decrease with the number of particles $K$ as we empirically observe in Section 6.1. We provide a detailed review of VIMCO in Appendix F.

The Reweighted Wake-Sleep (RWS) algorithm [20] is an extension of the original Wake-Sleep algorithm (ws) [19] that alternates between two distinct learning phases for optimizing importance weighted objectives. A detailed review of RWS and ws is available in Appendix F.

The Thermodynamic Variational Objective (TVO) [21] is a lower bound to $\log p_\theta(\mathbf{x})$ that stems from a Riemannian approximation of the Thermodynamic Variational Identity (TVI), and unifies the objectives of Variational Inference and Wake-Sleep. Evaluating the gradient involves differentiating

through an expectation over a distribution with an intractable normalizing constant. To accommodate this, the authors propose an estimator that generalizes the score function estimator based on a tractable covariance term. We review the TVO in more detail in Appendix F.

Given a deterministic *sampling path* $g(\boldsymbol{\epsilon}; \theta)$ such that $\mathbf{z} \sim p_\theta(\mathbf{z})$ and $\mathbf{z} = g(\boldsymbol{\epsilon}; \theta), \boldsymbol{\epsilon} \sim p(\boldsymbol{\epsilon})$ are equivalent, one can derive a *pathwise gradient estimator* of the form $\nabla_\theta \mathbb{E}_{p_\theta(\mathbf{z})}[f_\theta(\mathbf{z})] = \mathbb{E}_{p(\boldsymbol{\epsilon})}[\nabla_\theta f_\theta(g(\boldsymbol{\epsilon}; \theta))]$. This estimator – introduced in machine learning as the *reparameterization trick* or *stochastic backpropagation* [2, 3] – exhibits low variance thanks to the structural information provided by the sampling path. Notably, a zero expectation term can be removed from the estimator [32]. Extending on this, [24] derives an alternative gradient estimator for IWAE that exhibits $\mathrm{SNR} \sim K^{1/2}$, as opposed to $\mathrm{SNR} \sim K^{-1/2}$ for the *standard* IWAE objective [23].

Continuous relaxations of discrete distributions yield a biased low-variance gradient estimate thanks to the reparameterization trick [16, 33]. Discrete samples can be obtained using the Straight-Through estimator [5, 34]. The resulting gradient estimate remains biased, but can be used as a control variate for the score function objective, resulting in an unbiased low-variance estimate of the gradient [15, 35].

## 6 Experimental Results

We conduct a number of experiments[1] on benchmarks that have previously been used to test score function based estimators. All models are trained via stochastic gradient ascent using the Adam optimizer [36] with default parameters. We use regular gradients on the training objective for the generative model parameters $\theta$. The SNR for $\theta$ scales as $\mathcal{O}(K^{1/2})$ [23].

### 6.1 Asymptotic Variance

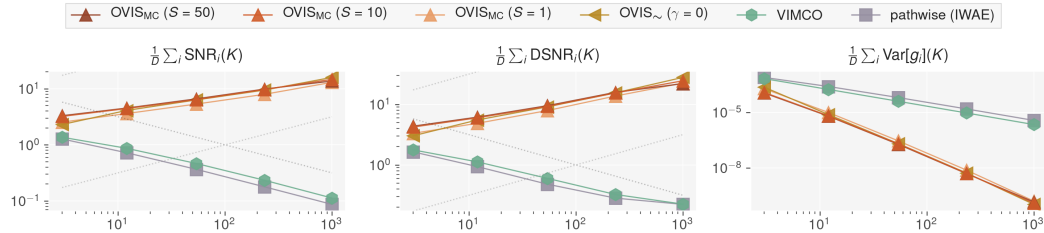

Figure 1: Gaussian model. Parameter-wise average of the asymptotic SNR, DSNR and variance of the gradients of the parameter $b$ for different number of particles $K \in [3, 1000]$ using $10^4$ MC samples. The dotted lines stand for $y = 10^{\pm 1} K^{\pm 0.5}$.

Following [23], we empirically corroborate the asymptotic properties of the OVIS gradient estimator by means of the following simple model:

$$\mathbf{z} \sim \mathcal{N}(\mathbf{z}; \boldsymbol{\mu}, \mathbf{I}), \quad \mathbf{x}|\mathbf{z} \sim \mathcal{N}(\mathbf{x}; \mathbf{z}, \mathbf{I}), \quad q_\phi(\mathbf{z}|\mathbf{x}) = \mathcal{N}\left(\mathbf{z}; \mathbf{A}\mathbf{x} + \mathbf{b}, \tfrac{2}{3}\mathbf{I}\right).$$

where $\mathbf{x}$ and $\mathbf{z}$ are real vectors of size $D = 20$. We sample $N = 1024$ points $\{\mathbf{x}^{(n)}\}_{n=1}^N$ from the *true* model where $\boldsymbol{\mu}^\star \sim \mathcal{N}(\mathbf{0}, \mathbf{I})$. The optimal parameters are $\mathbf{A}^\star = \mathbf{I}/2$, $\mathbf{b}^\star = \boldsymbol{\mu}^\star/2$, and $\boldsymbol{\mu}^\star = \frac{1}{N}\sum_{n=1}^N \mathbf{x}^{(n)}$. The model parameters are obtained by adding Gaussian noise of scale $\epsilon = 10^{-3}$. We measure the variance and the SNR of the gradients with $10^4$ MC samples. We also measured the *directional* SNR (DSNR [23]) to probe if our results hold in the multidimensional case.

In Figure 1 we report the gradient statistics for $\mathbf{b}$. We observe that using more samples in the standard IWAE leads to a decrease in SNR as $\mathcal{O}(K^{-1/2})$ for both VIMCO and the pathwise-IWAE [23]. The tighter variance control provided by OVIS leads the variance to decrease almost at a rate $\mathcal{O}(K^{-3})$, resulting in a measured SNR not far from $\mathcal{O}(K^{1/2})$ both for $\mathrm{OVIS_{MC}}$ and $\mathrm{OVIS_\sim}$. This shows that, despite the approximations, the proposed gradient estimators $\mathrm{OVIS_{MC}}$ and $\mathrm{OVIS_\sim}$ are capable of achieving the theoretical SNR of $\mathcal{O}(K^{1/2})$ derived in the asymptotic analysis in Section 3.

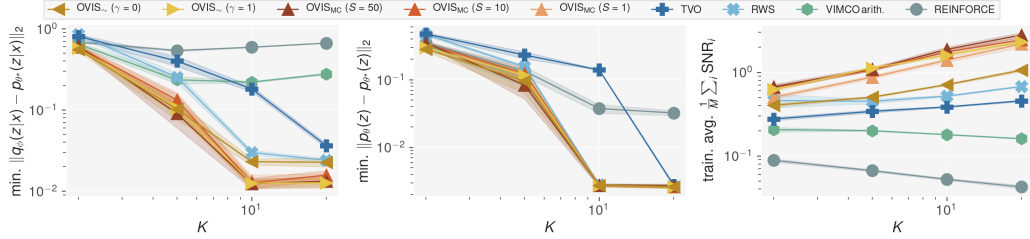

Figure 2: Training of the Gaussian mixture model. Minimum test-diagnostics recorded during training and training average of the SNR of the gradients of $\phi$ with $M = \text{card}(\phi)$. In contrast to VIMCO, OVIS$_\sim$ and OVIS$_{\text{MC}}$ all benefit from the increase of the particles budget, OVIS$_{\text{MC}}$ yields the most accurate posterior among the compared methods.

In Appendix G, we learn the parameters of the Gaussian model using OVIS, RWS, VIMCO and the TVO. We find that optimal variance reduction translates into a more accurate estimation of the optimal parameters of the inference network when compared to RWS, VIMCO and the TVO.

## 6.2 Gaussian Mixture Model

We evaluate OVIS on a Gaussian Mixture Model and show that, unlike VIMCO [9], our method yields better inference networks as the number of particles $K$ increases. Following [9], we define:

$$p_\theta(z) = \text{Cat}(z \,|\, \text{softmax}(\theta)) \quad p(x|z) = \mathcal{N}\left(x|\mu_z, \sigma_z^2\right) \quad q_\phi(z|x) = \text{Cat}\left(z \,|\, \text{softmax}\left(\eta_\phi(x)\right)\right)$$

where $z \in \{0, \dots, C-1\}$, $\mu_z = 10z$, $\sigma_z = 5$, and $C = 20$ is the number of clusters. The inference network $\eta_\phi$ is parameterized by a multilayer perceptron with architecture 1–16–$C$ and $\tanh$ activations. The true generative model is set to $p_{\theta^\star}(z = c) = (c + 5)/\sum_{i=1}^{C}(i + 5)$.

All models are trained for 100k steps with 5 random seeds. We compare OVIS with VIMCO, RWS with wake-$\phi$ update, Reinforce, and the TVO. For the latter we chose to use 5 partitions and $\beta_1 = 10^{-2}$, after a hyperparameter search over $\beta_1 \in \{10^{-1}, 10^{-1.5}, 10^{-2}, 10^{-2.5}, 10^{-3}\}$ and $\{2, 5\}$ partitions.

Each model is evaluated on a held-out test set of size $M = 100$. We measure the accuracy of the learned posterior $q_\phi(z|x)$ by its average $L_2$ distance from the true posterior, i.e. $\frac{1}{M}\sum_{m=1}^{M}\left\|q_\phi\left(z|x^{(m)}\right) - p_{\theta^\star}\left(z|x^{(m)}\right)\right\|_2$. As a sanity check, we assess the quality of the generative model using $\|\text{softmax}(\theta) - \text{softmax}(\theta^\star)\|_2$. The SNR of the gradients for the parameters $\phi$ is evaluated on one mini-batch of data using 500 MC samples.

We report our main results in Figure 2, and training curves in Appendix H. In contrast to VIMCO, the accuracy of the posteriors learned using OVIS$_{\text{MC}}$ and OVIS$_\sim$ all improve monotonically with $K$ and outperform the baseline estimators, independently of the choice of the number of auxiliary particles $S$. All OVIS methods outperform the state-of-the-art estimators RWS and the TVO, as measured by the $L_2$ distance between the approximate and the true posterior.

## 6.3 Deep Generative Models

We utilize the OVIS estimators to learn the parameters of both discrete and continuous deep generative models using stochastic gradient ascent. The base learning rate is fixed to $3 \cdot 10^{-4}$, we use mini-batches of size 24 and train all models for $4 \cdot 10^6$ steps. We use the statically binarized MNIST dataset [37] with the original training/validation/test splits of size 50k/10k/10k. We follow the experimental protocol as detailed in [21], including the $\beta$ partition for the TVO and the exact architecture of the models. We use a three-layer Sigmoid Belief Network [38] as an archetype of discrete generative model [13, 14, 21] and a Gaussian Variational Autoencoder [2] with 200 latent variables. All models are trained with three initial random seeds and for $K \in \{5, 10, 50\}$ particles.

We assess the performance based on the marginal log-likelihood estimate $\log \hat{p}_\theta(\mathbf{x}) = \mathcal{L}_{5000}(\mathbf{x})$, that we evaluate on 10k *training* data points, such as to disentangle the training dynamics from the regularisation effect that is specific to each method. We measure the quality of the inference network solution using the divergence $\mathcal{D}_{\text{KL}}\left(q_\phi(\mathbf{z}|\mathbf{x})||p_\theta(\mathbf{z}|\mathbf{x})\right) \approx \log \hat{p}_\theta(\mathbf{x}) - \mathcal{L}_1(\mathbf{x})$. The full training curves

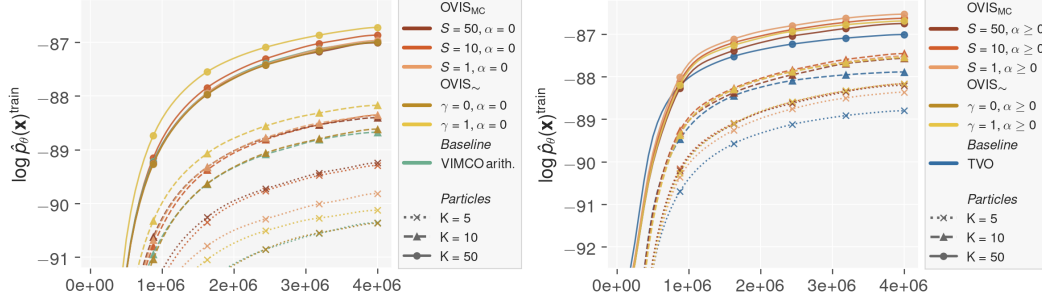

Figure 3: Training a Sigmoid Belief Network on Binarized MNIST. (Left) Optimizing for the importance weighted bound $\mathcal{L}_K$ using OVIS. (Right) Optimizing for the Rényi importance lower bound $\mathcal{L}_K^\alpha$ using OVIS with $\alpha$ annealing $0.99 \to 0$. The curves are averaged over three seeds and smoothed for clarity.

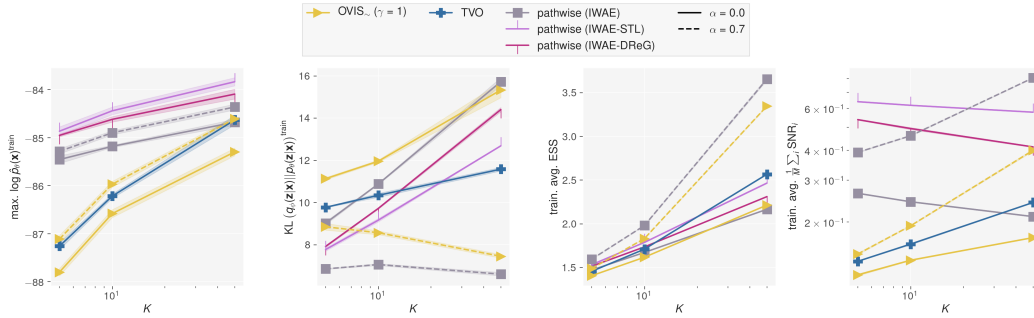

Figure 4: Training a one layer Gaussian VAE. Maximum recorded training $\log \hat{p}_\theta(\mathbf{x})$, final estimate of the bound $\mathcal{D}_{\mathrm{KL}}\left(q_\phi(\mathbf{z}|\mathbf{x})||p_\theta(\mathbf{z}|\mathbf{x})\right)$ and training average of the ESS and of the SNR. OVIS yields similar likelihood performances as the TVO but benefits from a tighter bound thanks to optimizing for the IWR bound.

– including the test log likelihood and divergences – are available in Appendix J. We will show that OVIS improves over VIMCO, on which it extends, and we show that combining OVIS$_\sim$ with the Variational Rényi bound (IWR) as described in Section 4 outperforms the TVO.

### 6.3.1 Sigmoid Belief Network (SBN)

**A. Comparison with VIMCO**   We learn the parameters of the SBN using the OVIS estimators for the IWAE bound and use VIMCO as a baseline. We report $\log \hat{p}_\theta(\mathbf{x})$ in the left plot of Figure 3. All OVIS methods outperform VIMCO, ergo supporting the advantage of optimal variance reduction. When using a small number of particles $K = 5$, learning can be greatly improved by using an accurate MC estimate of the optimal control variate, as suggested by OVIS$_{\mathrm{MC}}(S = 50)$ which allows gaining $+1.0$ nats over VIMCO. While OVIS($\gamma = 0$), designed for large ESS barely improved over VIMCO, the biased OVIS$_\sim(\gamma = 1)$ for low ESS performed significantly better than other methods for $K \geq 10$, which coincides with the ESS measured in the range $[1.0, 3.5]$ for all methods. We attribute the relative decrease of performances observed for OVIS$_{\mathrm{MC}}$ for $K = 50$ to *posterior collapse*.

**B. Training using IWR bounds**   In Figure 3 (right) we train the SBN using OVIS and the TVO. OVIS is coupled with the objective $\mathcal{L}_K^\alpha$ for which we anneal the parameter $\alpha$ from $0.99$ ($\mathcal{L}_K^{0.99} \approx \mathcal{L}_1$) to $0$ ($\mathcal{L}_K^0 = \mathcal{L}_K$) during $1e6$ steps using geometric interpolation. For all $K$ values, OVIS outperform the TVO and OVIS$_\sim(\gamma = 1)$ performs comparably with OVIS$_{\mathrm{MC}}$.

### 6.3.2 Gaussian Variational Autoencoder (VAE)

In Figure 4 we train the Gaussian VAE using the standard pathwise IWAE, Sticking the Landing (STL) [32], DReG [24], the TVO and OVIS$_\sim(\gamma = 1)$.

OVIS is applied to the IWR bound with $\alpha = 0.7$. As measured by the training likelihood, OVIS$_\sim(\gamma = 1)$ coupled with the IWR bound performs on par with the TVO, which bridges the gap to the standard pathwise IWAE for $K = 50$, although different objectives are at play. The advanced pathwise estimators (STL and DReG) outperform all other methods. Measuring the quality of the learned proposals $q_\phi(\mathbf{z}|\mathbf{x})$ using the KL divergence allows disentangling the TVO and OVIS$_\sim$ methods, as OVIS$(\gamma = 1)$ applied to the IWR bound outputs higher-quality approximate posteriors for all considered number of particles.

### 6.4 A final Note on OVIS$_\sim(\gamma = 1)$

OVIS$_\sim(\gamma = 1)$ generates training dynamics that are superior to the baseline TVO and to OVIS$_{\text{MC}}$ given a comparable particle budget (appendix I). We interpret this result as a consequence of the ESS-specific design, which also appeared to be robust to the choice of $\alpha$ in the IWR objective. This also corroborates the results of [32], that suppressing the term $-\sum_k v_k \mathbf{h}_k$ from the gradient estimate improves learning. We therefore recommend the practitioner to first experiment with OVIS$_\sim(\gamma = 1)$ since it delivers competitive results at a reasonable computational cost.

## 7 Conclusion

We proposed OVIS, a gradient estimator that is generally applicable to deep models with stochastic variables, and is empirically shown to have optimal variance control. This property is achieved by identifying and canceling terms in the estimator that solely contribute to the variance. We expect that in practice it will often be a good trade-off to use a looser bound with a higher effective sample size, e.g. by utilizing the OVIS estimator with the importance weighted Rényi bound, allowing control of this trade-off via an additional scalar smoothing parameter. This sentiment is supported by our method demonstrating better performance than the current state-of-the-art.

## 8 Financial Disclosure

The PhD program supporting Valentin Liévin is partially funded by Google. This research was supported by the NVIDIA Corporation with the donation of GPUs.

## 9 Broader Impact

This work proposes OVIS, an improvement to the score function gradient estimator in the form of optimal control variates for variance reduction. As briefly touched upon in the introduction, OVIS has potential practical use cases across several branches of machine learning. As such, the potential impact of this research is broad, and we will therefore limit the scope of this section to a few clear applications.

Improved inference over discrete spaces such as action spaces encountered within e.g. model-based reinforcement learning has the potential of reducing training time and result in more optimal behavior of the learning agent. This advancement has the capability to increase efficiency of e.g. autonomous robots used within manufacturing. Such progress is often coveted due to cost optimization, increased safety, and reduced manual labor for humans. However, as argued in [39], this development can also lead to immediate disadvantages such as worker displacement, potentially in terms of both tasks and geographic location.

Another probable avenue of impact of this research is within machine comprehension. A topic within this field is reading, with practical applications such as chatbots. This use of machine learning has seen rapid growth and commercial interest over recent years [40]. Apart from the clear consumer benefits of these bots, focus has also broadened to other cases of use for social benefits [41]. However, as with most other machine learning inventions, chatbots can be exploited for malicious purposes such as automated spread of misinformation, e.g. during elections [42].

As with other theoretical advances such as those presented in this paper, consequences are not immediate and depend on the applications in which the research is utilized. It is our hope that this research will ultimately be of practical use with a tangible positive impact.

## Footnotes

[1]The full experimental framework is available at github.com/vlievin/ovis

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
