[Supplementary Material]

## A Derivation of the Score Function Estimator

Given $K$ samples, the objective being maximized is

$$\mathcal{L}_K(\mathbf{x}) := \mathbb{E}\left[\log \hat{Z}\right] \qquad \hat{Z} := \frac{1}{K}\sum_{k=1}^{K} w_k \qquad w_k := \frac{p_\theta(\mathbf{x}, \mathbf{z}_k)}{q_\phi(\mathbf{z}_k|\mathbf{x})} \ . \tag{19}$$

The gradients of the multi-sample objective $\mathcal{L}_K$ with respect to the parameter $\phi$ can be expressed as a sum of two terms, one arising from the expectation over the variational posterior $q_\phi(\mathbf{z}_{1:K}|\mathbf{x}) := \prod_{k=1}^{K} q_\phi(\mathbf{z}_k|\mathbf{x})$ and one from $\log \hat{Z}$:

$$\nabla_\phi \mathcal{L}_K = \underbrace{\mathbb{E}\left[\log \hat{Z}\frac{\nabla_\phi q_\phi(\mathbf{z}_{1:K}|\mathbf{x})}{q_\phi(\mathbf{z}_{1:K}|\mathbf{x})}\right]}_{\text{(a)}} + \underbrace{\mathbb{E}\left[\nabla_\phi \log \hat{Z}\right]}_{\text{(b)}} \ .$$

The term **(a)** yields the traditional score function estimator

$$\begin{aligned}\text{(a)} &= \mathbb{E}\left[\log \hat{Z}\, \nabla_\phi \log q_\phi(\mathbf{z}_{1:K}|\mathbf{x})\right]\\ &= \mathbb{E}\left[\log \hat{Z}\sum_{k=1}^{K}\nabla_\phi \log q_\phi(\mathbf{z}_k|\mathbf{x})\right] \ .\end{aligned} \tag{20}$$

The term **(b)** is

$$\begin{aligned}\text{(b)} &= \mathbb{E}\left[\nabla_\phi \log \frac{1}{K}\sum_{k=1}^{K} w_k\right]\\ &= \mathbb{E}\left[\frac{1}{\frac{1}{K}\sum_{k=1}^{K} w_k}\nabla_\phi \frac{1}{K}\sum_{k=1}^{K} w_k\right]\\ &= \mathbb{E}\left[\frac{1}{\sum_{l=1}^{K} w_l}\sum_{k=1}^{K}\nabla_\phi w_k\right]\\ &= \mathbb{E}\left[\frac{1}{\sum_{l=1}^{K} w_l}\sum_{k=1}^{K} w_k\nabla_\phi \log w_k\right]\\ &= \mathbb{E}\left[\sum_{k=1}^{K} v_k\nabla_\phi \log w_k\right], \quad v_k = \frac{w_k}{\sum_{l=1}^{K} w_l}\\ &= -\mathbb{E}\left[\sum_{k=1}^{K} v_k\nabla_\phi \log q_\phi(\mathbf{z}_k|\mathbf{x})\right] \ .\end{aligned} \tag{21}$$

The derivation yields a factorized expression of the gradients

$$\nabla_\phi \mathcal{L}_K = \mathbb{E}_{q_\phi(\mathbf{z}_{1:K}|\mathbf{x})}\left[\sum_{k=1}^{K}\left(\log \hat{Z} - v_k\right)\mathbf{h}_k\right] \quad \text{with} \quad \mathbf{h}_k := \nabla_\phi \log q_\phi(\mathbf{z}_k|\mathbf{x}) \ . \tag{22}$$

## B Asymptotic Analysis

We present here a short derivation and direct the reader to [23] for the fine prints of the proof. The main requirement is that $w_k$ is bounded, so that $\hat{Z} - Z$ (with $Z = p(\mathbf{x})$) will converge to 0 almost surely as $K \to \infty$. We can also state this through the central limit theorem by noting that $\hat{Z} - Z = \frac{1}{K}\sum_k (w_k - Z)$ is the sum of $K$ independent terms so if $\text{Var}_1[w_1]$ is finite then $\hat{Z} - Z$ will converge to a Gaussian distribution with mean $\mathbb{E}[\hat{Z} - Z] = 0$ and variance $\text{Var}[\hat{Z} - Z] = \frac{1}{K}\text{Var}_1[w_1]$. The $K^{-1}$ factor on the variance follows from independence. This means that in a Taylor expansion in $\hat{Z} - Z$ higher order terms will be suppressed.

Rewriting $\mathbf{g}$ in terms of $\log \hat{Z}$:

$$\mathbf{g} = \sum_k (d_k - c_k) \mathbf{h}_k = \sum_k \left( \log \hat{Z} - w_k \frac{\partial}{\partial w_k} \log \hat{Z} - c_k \right) \mathbf{h}_k \tag{23}$$

and using the second-order Taylor expansion of $\log \hat{Z}$ about $Z$:

$$\log \hat{Z} \approx \log Z + \frac{\hat{Z} - Z}{Z} - \frac{(\hat{Z} - Z)^2}{2Z^2} \tag{24}$$

we have

$$\log \hat{Z} \approx \log Z - \frac{3}{2} + \frac{2}{KZ} \sum_l w_l - \frac{1}{2K^2 Z^2} \left( \sum_l w_l \right)^2 \tag{25}$$

$$\frac{\partial}{\partial w_k} \log \hat{Z} \approx \frac{2}{KZ} - \frac{1}{K^2 Z^2} \sum_l w_l . \tag{26}$$

The term $d_k$ can thus be approximated as follows:

$$d_k = \log \hat{Z} - w_k \frac{\partial}{\partial w_k} \log \hat{Z}$$

$$\approx \log Z - \frac{3}{2} + \frac{2}{KZ} \sum_{l \neq k} w_l - \frac{1}{2K^2 Z^2} \left( \sum_l w_l \right)^2 + \frac{1}{K^2 Z^2} w_k \sum_l w_l$$

$$= \log Z - \frac{3}{2} + \frac{2}{KZ} \sum_{l \neq k} w_l - \frac{1}{2K^2 Z^2} \left( \sum_{l \neq k} w_l \right)^2 + \frac{1}{2K^2 Z^2} w_k^2 \tag{27}$$

where we used

$$\left( \sum_l w_l \right)^2 = \left( \sum_{l \neq k} w_l \right)^2 + w_k^2 + 2 w_k \sum_{l \neq k} w_l .$$

By separately collecting the terms that depend and do not depend on $\mathbf{z}_k$ into $f_k = f_k(\mathbf{z}_k, \mathbf{z}_{-k})$ and $f_{-k} = f_{-k}(\mathbf{z}_{-k})$, respectively, we can rewrite the estimator $\mathbf{g}$ as:

$$\mathbf{g} = \sum_k (f_k + f_{-k} - c_k) \mathbf{h}_k \tag{28}$$

and from (27) we have

$$f_k \approx \frac{w_k^2}{2K^2 Z^2} \tag{29}$$

$$f_{-k} \approx \log Z - \frac{3}{2} + \frac{2}{KZ} \sum_{l \neq k} w_l - \frac{1}{2K^2 Z^2} \left( \sum_{l \neq k} w_l \right)^2 . \tag{30}$$

## C  Asymptotic Expectation and Variance

We derive here the asymptotic expectation and variance of the gradient estimator $\mathbf{g}$ in the limit $K \to \infty$.

### C.1  Expectation

If both $f_{-k}$ and $c_k$ are independent of $\mathbf{z}_k$, we can write:

$$\mathbb{E}[\mathbf{g}] = \mathbb{E} \left[ \sum_k (f_k + f_{-k} - c_k) \mathbf{h}_k \right] = \sum_k \mathbb{E} [f_k \mathbf{h}_k] \tag{31}$$

where we used that $\mathbb{E}\left[f_{-k}\mathbf{h}_k\right]$ and $\mathbb{E}\left[c_k\mathbf{h}_k\right]$ are zero. In the limit $K \to \infty$, each term of the sum can be expanded with the approximation (29) and simplified:

$$\mathbb{E}\left[f_k\mathbf{h}_k\right] \approx \mathbb{E}\left[\frac{w_k^2}{2K^2Z^2}\mathbf{h}_k\right] = \frac{1}{2K^2Z^2}\mathbb{E}_1\left[w_1^2\mathbf{h}_1\right] \tag{32}$$

where $\mathbb{E}_1$ denotes an expectation over the posterior $q_\phi(\mathbf{z}_1|\mathbf{x})$. The last step follows from the fact that the latent variables $\{\mathbf{z}_k\}_{k=1}^K$ are i.i.d. and the argument of the expectation only depends on one of them. In conclusion, the expectation is:

$$\mathbb{E}[\mathbf{g}] = \sum_k \mathbb{E}\left[f_k\mathbf{h}_k\right] \approx \frac{1}{2KZ^2}\mathbb{E}_1\left[w_1^2\mathbf{h}_1\right] = \mathcal{O}(K^{-1}) \tag{33}$$

irrespective of $f_{-k}$ and $c_k$.

### C.2 Variance

If $c_k$ is chosen to be $c_k(\mathbf{z}_{-k}) = f_{-k}(\mathbf{z}_{-k})$ then we can again use the approximation (29) for $K \to \infty$ and get the asymptotic variance:

$$\text{Var}[\mathbf{g}] = \text{Var}\left[\sum_k f_k\mathbf{h}_k\right] \tag{34}$$

$$\approx \text{Var}\left[\sum_k \frac{w_k^2}{2K^2Z^2}\mathbf{h}_k\right] \tag{35}$$

$$= \frac{1}{4K^4Z^4}\sum_k \text{Var}_k\left[w_k^2\mathbf{h}_k\right] \tag{36}$$

$$= \frac{1}{4K^3Z^4}\text{Var}_1\left[w_1^2\mathbf{h}_1\right] \tag{37}$$

$$= \mathcal{O}(K^{-3}) \tag{38}$$

where $\text{Var}_k$ denotes the variance over the $k$th approximate posterior $q_\phi(\mathbf{z}_k|\mathbf{x})$, and we used the fact that the latent variables $\{\mathbf{z}_k\}_{k=1}^K$ are i.i.d. and therefore there are no covariance terms.

## D  Optimal Control for the ESS Limits and Unified Interpolation

### D.1  Control Variate for Large ESS

In the gradient estimator $\mathbf{g} = \sum_k \left(\log \hat{Z} - \frac{\partial \log \hat{Z}}{\partial w_k}w_k - c_k\right)\mathbf{h}_k$, we consider the $k$th term in the sum, where we have that $\hat{Z} - \widetilde{Z}_{[-k]} = \frac{w_k}{K} \to 0$ as $K \to \infty$. We can therefore expand $\log \hat{Z}$ as a Taylor series around $\hat{Z} = \widetilde{Z}_{[-k]}$, obtaining:

$$\log \hat{Z} = \log \widetilde{Z}_{[-k]} + \sum_{p=1}^{\infty} \frac{(-1)^{p+1}}{p}\left(\frac{w_k}{K\widetilde{Z}_{[-k]}}\right)^p \tag{39}$$

$$\frac{\partial \log \hat{Z}}{\partial w_k} = \frac{1}{w_k}\sum_{p=1}^{\infty}(-1)^{p+1}\left(\frac{w_k}{K\widetilde{Z}_{[-k]}}\right)^p. \tag{40}$$

Inserting these results into the gradient estimator and using the expression $\mathbf{g} = \sum_k (f_k + f_{-k} - c_k)\mathbf{h}_k$ we see that

$$f_{-k} = \log \widetilde{Z}_{[-k]} \tag{41}$$

$$f_k = \sum_{p=1}^{\infty}(-1)^{p+1}\left(\frac{1}{p} - 1\right)\left(\frac{w_k}{K\widetilde{Z}_{[-k]}}\right)^p \tag{42}$$

$$= \sum_{p=2}^{\infty}(-1)^p\left(1 - \frac{1}{p}\right)\left(\frac{w_k}{K\widetilde{Z}_{[-k]}}\right)^p. \tag{43}$$

We now use this to simplify the optimal control variate (10) to leading order. Since $f_k$ is order $K^{-2}$, the term $\mathbb{E}_k\left[f_k\|\mathbf{h}_k\|^2\right]$ will be of order $K^{-2}$ as well. The $l \neq k$ terms $\mathbb{E}_k\left[f_l\mathbf{h}_k^T\mathbf{h}_l\right]$ get non-zero contributions only through the $w_k$ term in $f_l$. As $w_k$ appears in $\widetilde{Z}_{[-l]}$ with a prefactor $K^{-1}$, we have $\mathbb{E}_k\left[f_l\mathbf{h}_k^T\mathbf{h}_l\right] = \mathcal{O}(K^{-3})$ for $l \neq k$, and the sum of these terms is $\mathcal{O}(K^{-2})$. Overall, this means that the second term in the control variate only gives a contribution of $\mathcal{O}(K^{-2})$ and thus can be ignored:

$$c_k \approx \log \widetilde{Z}_{[-k]} = \log \frac{1}{K}\sum_{l \neq k} w_l = \log \frac{1}{K-1}\sum_{l \neq k} w_l + \log(1 - \frac{1}{K}) \,. \tag{44}$$

Note that in the simplifying approximation in Section 4 we argue that the $l \neq k$ terms $\mathbb{E}_k\left[f_l\mathbf{h}_k^T\mathbf{h}_l\right]$ can be omitted and only the $l = k$ term retained. Here we show that their overall contribution is the same order as the $l = k$ term. These results are not in contradiction because here we are only discussing orders and not the size of terms.

## D.2  Control Variate for Small ESS

In the case ESS $\approx 1$ we can write $\log \hat{Z}$ as a sum of two terms:

$$\log \hat{Z} = \log \frac{w_{k'}}{K} + \log\left(1 + \frac{K\tilde{Z}_{[-k']}}{w_{k'}}\right) \,, \tag{45}$$

where $w_{k'}$ is the dominating weight. The first term dominates and the second can be ignored to leading order. We will leave out a derivation for non-leading terms for brevity. So the gradient estimator $\mathbf{g} = \sum_k \left(\log \hat{Z} - \frac{\partial \log \hat{Z}}{\partial w_k}w_k - c_k\right)\mathbf{h}_k$ simply becomes $\mathbf{g} \approx \sum_k \left(\log \frac{w_{k'}}{K} - \delta_{k,k'} - c_k\right)\mathbf{h}_k$. This corresponds to $f_k = \delta_{k,k'}\log w_{k'}$ and $f_{-k} = (1 - \delta_{k,k'})\log w_{k'} - \delta_{k,k'} - \log K$. Inserting this into Equation (11) we get:

$$\mathbf{g} = \sum_k\left(f_k - \sum_l \frac{\mathbb{E}_k\left[f_l\mathbf{h}_k^T\mathbf{h}_l\right]}{\mathbb{E}_k\left[\|\mathbf{h}_k\|^2\right]}\right)\mathbf{h}_k = \left(\log w_{k'} - \frac{\mathbb{E}_{k'}\left[\log w_{k'}\|\mathbf{h}_{k'}\|^2\right]}{\mathbb{E}_{k'}\left[\|\mathbf{h}_{k'}\|^2\right]}\right)\mathbf{h}_{k'} \,. \tag{46}$$

Estimating the expectation $\mathbb{E}_{k'}[\dots]$ in Equation (46) using i.i.d. samples from $q_\phi(\mathbf{z}|\mathbf{x})$ is computationally involved. Therefore we resort to the approximation $\mathbf{g} \approx \sum_k\left(\log \frac{w_{k'}}{K} - \delta_{k,k'} - c_k\right)\mathbf{h}_k$ and $\delta_{k,k'} \approx v_k$, which holds in the limit ESS $\to 1$. We get:

$$c_k \approx \log \hat{Z}_{[-k]} - v_k = \log \frac{1}{K-1}\sum_{l \neq k} w_l - v_k \,. \tag{47}$$

Relying on the approximation $\delta_{k,k'} \approx v_k$ corresponds to suppressing the term $-v_k$ of the prefactors $d_k = \log \hat{Z} - v_k$ and does not guarantee the resulting objective to be unbiased for ESS $> 1$. Suppressing this term has been explored in depth for the pathwise gradient estimator [32]. The gradient estimator $\sum_k v_k\mathbf{h}_k$ corresponds to *wake-phase* update in RWS.

## D.3  Unified Interpolation

We unify the two ESS limits under a unifying expression OVIS$_\sim$ defined for a scalar $\gamma \in [0, 1]$:

$$c_k^\gamma := \log \hat{Z}_{[-k]} - \gamma v_k + (1 - \gamma)\log(1 - 1/K) \tag{48}$$

where

$$c_k^0 = \log \frac{1}{K-1}\sum_{l \neq k} w_l + \log(1 - 1/K) \tag{49}$$

$$c_k^1 = \log \frac{1}{K-1}\sum_{l \neq k} w_l - v_k \,. \tag{50}$$

# E    Rényi Importance Weighted Bound

All the analysis applied to the score function estimator for the importance weighted bound including asymptotic SNR can directly be carried over to the Rényi importance weighted bound $\mathcal{L}_K^\alpha(\mathbf{x})$ because all the independence properties are unchanged. The score function estimator of the gradient of $\phi$ is given by

$$\nabla_\phi \mathcal{L}_K^\alpha(\mathbf{x}) = \sum_k \left( \frac{1}{1-\alpha} \log \hat{Z}(\alpha) - v_k(\alpha) \right) \mathbf{h}_k, \qquad v_k(\alpha) = \frac{w_k^{1-\alpha}}{\sum_l w_l^{1-\alpha}} . \qquad (51)$$

The OVIS$_{\text{MC}}$ formulation holds using $d_k = \frac{1}{1-\alpha} \log \hat{Z}(\alpha) - v_k(\alpha)$ within the equation 12. Similarly for the asymptotic expression OVIS$_\sim$, the unified control variate 17 becomes:

$$c_k^\gamma := \log \frac{1}{1-\alpha} \log \hat{Z}_{[-k]}(\alpha) - \gamma v_k + (1-\gamma) \log(1 - 1/K) \qquad (52)$$

# F    Gradient Estimators Review

In this paper, gradient *ascent* is considered (i.e. maximizing the objective function). The expression of the gradient estimators presented below are therefore adapted for this setting.

**VIMCO**    The formulation of the VIMCO [14] control variate exploits the structure of $\hat{Z} := \frac{1}{K} \sum_l w_l$ using $c_k := c_k(\mathbf{z}_{-k}) = \log \frac{1}{K} \sum_{l \neq k} w_l + \hat{w}_{[-k]}$ where $\hat{w}_{[-k]}$ stands for the arithmetic or geometric average of the weights $w_l$ given the set of outer samples $\mathbf{z}_{-k}$. Defining $\log \hat{Z}_{[-k]} := c_k$, the VIMCO estimator of the gradients is

$$\nabla_\phi \mathcal{L}_K = \mathbb{E}_{q_\phi(\mathbf{z}_{1:K}|\mathbf{x})} \left[ \underbrace{\sum_{k=1}^K \left( \log \hat{Z} - \log \hat{Z}_{[-k]} \right) \mathbf{h}_k}_{(a)} + \underbrace{\sum_{k=1}^K v_k \nabla_\phi \log w_k}_{(b)} \right] . \qquad (53)$$

We refer to [14] for the derivation. Here, the term $\hat{Z}_{[-k]}$ can be expressed using the arithmetic and the geometric averaging [14]. The leave-one-sample estimate can be expressed as

$$\hat{Z}_{[-k]} = \frac{1}{K} \sum_{l \neq k} w_l + \hat{w}_{[-k]} \text{ with } \begin{cases} \hat{w}_{[-k]} = \frac{1}{K-1} \sum_{l \neq k} w_l & \text{(arithmetic)} \\ \hat{w}_{[-k]} = \exp \frac{1}{K-1} \sum_{l \neq k} \log w_l & \text{(geometric)} \end{cases} \qquad (54)$$

The term **(b)** is well-behaved because it is a convex combination of the K gradients $\nabla_\phi \log w_k$. However, the term **(a)** may dominate the term **(b)**. In contrast to VIMCO, OVIS allows controlling the variance of both terms **(a)** and **(b)**, resulting in a more optimal variance reduction. In the Reweighted Wake Sleep (RWS) with wake-wake-$\phi$ update, the gradient of the parameters $\phi$ of the inference network corresponds to the negative of the term **(b)**.

**Wake-sleep**    The algorithm [19] relies on two separate learning steps that are alternated during training: the *wake-phase* that updates the parameters of the generative model $\theta$ and the *sleep-phase* used to update the parameters of the inference network with parameters $\phi$. During the *wake-phase*, the generative model is optimized to maximize the evidence lower bound $\mathcal{L}_1$ given a set of observation $\mathbf{x} \sim p(\mathbf{x})$. During the *sleep-phase*, a set of observations and latent samples are *dreamed* from the model: $\mathbf{x}, \mathbf{z} \sim p_\theta(\mathbf{x}, \mathbf{z})$ and the parameters $\phi$ of the inference network are optimized to minimize the KL divergence between the true posterior of the generative model and the approximate posterior: $\mathcal{D}_{\text{KL}}\left(p_\theta(\mathbf{z}|\mathbf{x}) || q_\phi(\mathbf{z}|\mathbf{x})\right)$.

**Reweighted Wake-Sleep (RWS)**    extends the original Wake-Sleep algorithm for importance weighted objectives [20]. The generative model is now optimized for the importance weighted bound $\mathcal{L}_K$, which gives the following gradients

$$\nabla_\theta \mathcal{L}_K = \mathbb{E}_{q_\phi(\mathbf{z}_{1:K}|\mathbf{x})} \left[ \sum_k v_k \nabla_\theta \log w_k \right] \quad \text{(wake-phase } \theta\text{)} . \qquad (55)$$

The parameters $\phi$ of the inference network are optimized given two updates: the *sleep-phase* $\phi$ an the *wake-phase* $\phi$. The *sleep-phase* $\phi$ is identical to the original Wake-Sleep algorithm, the gradients of the parameters $\phi$ of the inference model are given by

$$-\nabla_\phi \mathbb{E}_{p_\theta(\mathbf{x})} \left[ \mathcal{D}_{\mathrm{KL}} \left( p_\theta(\mathbf{z}_{1:K}|\mathbf{x}) || q_\phi(\mathbf{z}_{1:K}|\mathbf{x}) \right) \right] = \mathbb{E}_{p_\theta(\mathbf{z}_{1:K},\mathbf{x})} \left[ \sum_k \mathbf{h}_k \right] \quad \text{(sleep-phase } \phi) . \quad (56)$$

The *wake-phase* $\phi$ differs from the original Wake-Sleep algorithm that samples $\mathbf{x}, \mathbf{z}$ are sampled respectively from the dataset and from the inference model $q_\phi(\mathbf{z}|\mathbf{x})$. In this cases the gradients are given by:

$$-\nabla_\phi \mathbb{E}_{p(\mathbf{x})} \left[ \mathcal{D}_{\mathrm{KL}} \left( p_\theta(\mathbf{z}_{1:K}|\mathbf{x}) || q_\phi(\mathbf{z}_{1:K}|\mathbf{x}) \right) \right] = \mathbb{E}_{p(\mathbf{x})} \left[ \mathbb{E}_{q_\phi(\mathbf{z}_{1:K}|\mathbf{x})} \left[ \sum_k v_k \mathbf{h}_k \right] \right] \quad \text{(wake-phase } \phi) . \quad (57)$$

Critically, in Variational Autoencoders one optimizes a lower bound of the marginal log-likelihood ($\mathcal{L}_K$), while RWS instead optimizes a biased estimate of the marginal log-likelihood $\log p(\mathbf{x})$. However, the bias decreases with $K$ [20]. [9] shows that RWS is a method of choice for training deep generative models and stochastic control flows. In particular, [9] shows that increasing the budget of particles $K$ benefits the learning of the inference network when using the wake-phase update (Wake-Wake algorithm).

We refer the reader to [20] for the derivations of the gradients and [9] for an extended review of the RWS algorithms for the training of deep generative models.

**The Thermodynamic Variational Objective (TVO)**  The gradient estimator consists of expressing the marginal log-likelihood $\log p_\theta(\mathbf{x})$ using Thermodynamic Integration (TI). Given two unnormalized densities $\tilde{\pi}_0(\mathbf{z})$ and $\tilde{\pi}_1(\mathbf{z})$ and their respective normalizing constants $Z_0, Z_1$ with $Z_i = \int \tilde{\pi}_i(\mathbf{z}) d\mathbf{z}$ given the unnormalized density $\tilde{\pi}_\beta(\mathbf{z}) := \pi_1(\mathbf{z})^\beta \pi_0^{1-\beta}(\mathbf{z})$ parameterized by $\beta \in [0,1]$, and the corresponding normalized density $\pi_\beta(\mathbf{z}) = \tilde{\pi}_\beta(\mathbf{z}) / \int \tilde{\pi}_\beta(\mathbf{z}) d\mathbf{z}$, TI seeks to evaluate the ratio of the normalizing constants using the identity

$$\log Z_1 - \log Z_0 = \int_0^1 \mathbb{E}_{\pi_\beta} \left[ \frac{d \log \tilde{\pi}_\beta(\mathbf{z})}{d\beta} \right] d\beta . \quad (58)$$

[21] connects TI to Variational Inference by setting the base densities as $\tilde{\pi}_0(\mathbf{z}) = q_\phi(\mathbf{z}|\mathbf{x})$ and $\tilde{\pi}_1(\mathbf{z}) = p_\theta(\mathbf{x}, \mathbf{z})$, which gives the Thermodynamic Variational Identity (TVI):

$$\log p_\theta(\mathbf{x}) = \int_0^1 \mathbb{E}_{\pi_\beta} \left[ \log \frac{p_\theta(\mathbf{x}, \mathbf{z})}{q_\phi(\mathbf{z}|\mathbf{x})} \right] d\beta. \quad (59)$$

Applying left Riemannian approximation yields the Thermodynamic Variational Objective (TVO):

$$\mathrm{TVO}(\theta, \phi, \mathbf{x}) = \frac{1}{P} \left[ \mathrm{ELBO}(\theta, \phi, \mathbf{x}) + \sum_{p=1}^{P-1} \mathbb{E}_{\pi_{\beta_P}} \left[ \log \frac{p_\theta(\mathbf{x}, \mathbf{z})}{q_\phi(\mathbf{z}|\mathbf{x})} \right] \right] \leq \log p_\theta(\mathbf{x}) . \quad (60)$$

Notably, the integrand $\mathbb{E}_{\pi_\beta} \left[ \log \frac{p_\theta(\mathbf{x},\mathbf{z})}{q_\phi(\mathbf{z}|\mathbf{x})} \right]$ is monotically increasing, which implies that the TVO is a lower-bound of the marginal log-likelihood.

The TVO allows connecting both Variational Inference and the Wake-Sleep objectives by observing that when using a partition of size $P = 1$, the left Riemannian approximation of the TVI, $\mathrm{TVO}_1^L(\theta, \phi, \mathbf{x}) = \mathrm{ELBO}(\theta, \phi, \mathbf{x})$ and the right Riemannian approximation of the TVI, $\mathrm{TVO}_1^U(\theta, \phi, \mathbf{x})$ is an upper bound of the marginal log-likelihood and equals the objective being maximized in the *wake-phase* for the parameters $\phi$ of the inference network.

Estimating the gradients of the TVO requires computing the gradient for each of the $P$ expectations $\mathbb{E}_{\pi_{\lambda,\beta}} [f_\lambda(\mathbf{z})]$ with respect to a parameter $\lambda := \{\theta, \phi\}$ where $f_\lambda(\mathbf{z}) = \log \frac{p_\theta(\mathbf{x},\mathbf{z})}{q_\phi(\mathbf{z}|\mathbf{x})}$ and $\mathbf{x}$ is fixed. In

the general case, differentiation through the expectation is not trivial. Therefore the authors propose a score function estimator

$$\nabla_\lambda \mathbb{E}_{\pi_{\lambda,\beta}} [f_\lambda(\mathbf{z})] = \mathbb{E}_{\pi_{\lambda,\beta}} [\nabla_\lambda f_\lambda(\mathbf{z})] + \mathrm{Cov}_{\pi_{\lambda,\beta}} [\nabla_\lambda \log \tilde{\pi}_{\lambda,\beta}(\mathbf{z}), f_\lambda(\mathbf{z})] \ , \tag{61}$$

where the covariance term can be expressed as

$$\mathbb{E}_{\pi_{\lambda,\beta}} \left[ \left( f_\lambda(\mathbf{z}) - \mathbb{E}_{\pi_{\lambda,\beta}} [f_\lambda(\mathbf{z})] \right) \left( \nabla_\lambda \log \tilde{\pi}_{\lambda,\beta}(\mathbf{z}) - \mathbb{E}_{\pi_{\lambda,\beta}} [\nabla_\lambda \log \tilde{\pi}_{\lambda,\beta}(\mathbf{z})] \right) \right] \ . \tag{62}$$

The covariance term arises when differentiating an expectation taken over a distribution with an intractable normalizing constant, such as $\pi_\beta(\mathbf{z})$ in the TVO. The normalizing constant can be substituted out, resulting in a covariance term involving the tractable un-normalized density $\tilde{\pi}_\beta(\mathbf{z})$. Hence, such a covariance term does not usually arise in IWAE due to the derivative of $q_\phi(\mathbf{z}|\mathbf{x})$ being available in closed form.

## G  Gaussian Model

Figure 5: Distribution of the gradients for an arbitrarily chosen component of the parameter $\mathbf{b}$. The tight control of the variance provided by OVIS allows keeping the distribution of gradients off-center.

**Distribution of gradients**   We report the distributions of the $10^4$ MC estimates of the gradient of the first component $b_0$ of the parameter $\mathbf{b}$. Figure 5. The pathwise estimator and VIMCO yield estimates which distributions are progressively centered around zero as $K \to \infty$. The faster decrease of the variance of the gradient estimate for OVIS results in a distribution of gradients that remains off-centered.

Figure 6: Asymptotic analysis of the gradients for OVIS$_\sim$ and the STL and DReG IWAE estimators.

**Analysis for advanced pathwise IWAE estimators**   We perform the experiment 3 using additional pathwise estimators: STL [32] and DReG-IWAE [24]. Both the STL and OVIS$_\sim(\gamma = 1)$ rely on the suppression of the term $\sum_k v_k \mathbf{h}_k$ from the gradient estimate and adopt the same behaviour: the variance decreases at a slower rate than OVIS$_\sim(\gamma = 0)$ and DReG, however, its bias remains constant as K is increased.

**Fitting the Gaussian Model**

Figure 7: Fitting the Gaussian toy model from section 6.1 and measuring the $\mathcal{L}_2$ distance with the optimal parameters as well as the variance and the SNR of the gradient estimates. OVIS methods target the optimal parameters $A^\star$ of the inference network more accurately than the baseline methods.

We study the relative effect of the different estimators when training the Gaussian toy model from section 6.1. The model is trained for 5.000 epochs using the Adam optimizer with a base learning rate of $10^{-3}$ and with a batch-size of 100. In Figure 7, we report the $L_2$ distance from the model parameters $A$ to the optimal parameters $A^\star$, the parameters-average SNR and parameters-average variance of the inference network ($\phi = \{A, \mathbf{b}\}$, $M = \mathrm{card}(\phi)$). We compare OVIS methods with VIMCO, the pathwise IWAE, RWS and the TVO for which we picked a partition size $P = 5$ and $\beta_1 = 10^{-3}$, although no extensive grid search has been implemented to identify the optimal choice for this parameters.

OVIS yields gradient estimates of lower variance than the other methods. The inference network solutions given by OVIS are slightly more accurate than the baseline methods RWS and the TVO, despite being slower to converge. OVIS, RWS and the TVO exhibit gradients with comparable SNR values, which indicate OVIS yield estimate of lower expected value, thus leading to a smaller maximum optimization step-size. Setting $\gamma = 0$ for OVIS$_\sim$ results in more accurate solutions than using $\gamma = 1$, this coincides with the measured ESS $\approx K$.

# H   Gaussian Mixture Model

Figure 8: Training curves for the Gaussian Mixture Model for different numbers of particles $K = [2, 5, 10, 20]$ samples averaged over 5 random seeds. The SNR is measured on one mini-batch and averaged over the $M$ parameters of the inference network. In contrast to VIMCO, OVIS estimators all generate gradients with a higher SNR. This results in a more accurate estimate of the true posterior, when compared to VIMCO and the baselines RWS and the TVO.

# I   Comparison of $\mathrm{OVIS}_\sim$ and $\mathrm{OVIS}_{\mathrm{MC}}$ with under a fixed Particle Budget

$\mathrm{OVIS}_{\mathrm{MC}}$ has complexity requires $K + S$ importance weights whereas $\mathrm{OVIS}_\sim$ requires only $K$. Estimating $\phi$ using $\mathrm{OVIS}_{\mathrm{MC}}$ requires a budget of $K' = K + S$ particles. The ratio $S/K$ is a trade-off between the tightness of the bound $\mathcal{L}_K$ and the variance of the control variate estimate. In the main text, we focus on studying the sole effect of the control variate given the bound $\mathcal{L}_K$. This corresponds to a sub-optimal use of the budget $K'$ because $\mathcal{L}_{K'}$ is tighter than $\mathcal{L}_K$. By contrast with the previous experiments, we trained the Gaussian VAE using the budget $K'$ optimally (i.e. relying on $\mathcal{L}_{K'}$ whenever no auxiliary samples are used). We observed that $\mathrm{OVIS}_\sim(\gamma = 1)$ outperforms $\mathrm{OVIS}_{\mathrm{MC}}$ despite the generative model is evaluated using $\mathcal{L}_{K'}$ in all cases (figure 9). This experiment will be detailed in the Appendix.

Figure 9: Training the Gaussian VAE model with a fixed and optimally used particle budget $K' = K + S$ and $\alpha = 0.7$.

# J   Training Curves for the Deep Generative Models

## J.1   Sigmoid Belief Network

Figure 10: Training curves for the Sigmoid Belief Network using $K = [5, 10, 50]$ particles, using two initial random seeds, with and without using the IWR bound. The number of active units is evaluated as $\text{AU} = \sum_{d=1}^{D} \mathbb{1} \left\{ \text{Cov}_{p(\mathbf{x})} \left( \mathbb{E}_{q_\phi(\mathbf{z}|\mathbf{x})} [\mathbf{z}_d] \right) \geq 0.01 \right\}$ [22] using 1000 MC samples for each element of a randomly sampled subset of 1000 data points. Warming up the model by optimizing for the IWR bound with $\alpha > 0$ allows activating a larger number of units and results in models scoring higher training likelihoods.

## J.2 Gaussian Variational Autoencoder

Figure 11: Training curves a Gaussian VAE using $K = [5, 10, 50]$ particles and using two initial random seeds. The OVIS estimators are used in tandem with the IWR bound with $\alpha$ fixed to $0.3$. OVIS for the IWR bound yields high-quality inference networks, as measured by the divergence $\mathcal{D}_{\mathrm{KL}}\left(p_\theta(\mathbf{z}|\mathbf{x})||q_\phi(\mathbf{z}|\mathbf{x})\right)$.

# K   Implementation Details for $\mathrm{OVIS}_\sim$

In order to save computational resources for large $K$ values, we implement the following factorization

$$\log \hat{Z} - \log \hat{Z}_{[-k]} = \log \frac{1 - 1/K}{1 - v_k} \ . \tag{63}$$

In order to guarantee computational stability, we clip the normalized importance weights $v_k$ using the default PyTorch value $\epsilon = 1.19e^{-7}$. The resulting gradient estimate, used in the main experiments, is

$$\mathbf{g} := \sum_k \left( \log \frac{1 - 1/K}{1 - \min(1 - \epsilon, v_k)} + (\gamma - 1)v_k - (1 - \gamma)\log(1 - 1/K) \right) \mathbf{h}_k \ . \tag{64}$$

Clipping the normalized importance weights can be interpreted as an instance of truncated importance sampling. Hence, the value of $\epsilon$ must be carefully selected. In the figure 12, we present a comparison of $\mathrm{OVIS}_\sim$ with and without clipping. The experiments indicate that the difference is insignificant when using the default $\epsilon$.

Figure 12: Effect of the importance weight clipping. Training the Gaussian Mixture Model, Sigmoid Belief Network and Gaussian VAE with and without clipping.