[Reviews · NeurIPS 2020]

Review 1

Summary and Contributions: The importance-weighted autoencoder (IWAE) has the well-known drawback that the Monte-Carlo approximation of its gradient for the variational parameters has a signal-to-noise ratio (SNR) which vanishes with the number of Monte Carlo samples, $K$. This is particularly concerning because IWAEs are motivated by the fact that they can remove the bias of conventional variational autoencoders in the limit as $K$ goes to infinity. A general solution to this problem was previously only available in reparametrisable models (i.e. not for discrete latent variables). This work proposes a control-variate construction -- derived through Taylor-series expansions -- which demonstrably stabilises the SNR and which is applicable even in non-reparametrisable models.

Strengths: - This work allows IWAEs (to be applied relatively easily in non-reparametrisable models which was hitherto difficult. - The proposed control-variate constructions are generic and require much less application-specific tuning than previously proposed control-variate approaches. - The computational cost of the proposed method is not much larger than for standard IWAE (the computational complexity of the "OVIS_MC" estimator from Eq. 12 seems to be $O(SK)$ which could quickly become prohibitive for larger numbers of additional samples $S$ and larger $K$; however, the simulations indicate that using $S=1$ additional samples is sufficient to avoid the breakdown normally suffered by IWAE).

Weaknesses: - I wonder how crucial the annealing scheme from the last paragraph in Section 4 is. Especially when $\alpha$ is not decreased to $0$ I imagine this could induce a bias which may be so large that it outweighs the bias reductions attained by using IWAE in the first place. - The only other weakness is related to the clarity of the exposition, especially around the "OVIS_~" estimator (see further details below). ==== EDIT: 2020-08-24 ===== replaced "$\alpha$ is not increased to $1$" by "$\alpha$ is not decreased to $0$" as I had had a typo in this part of my review

Correctness: - The work seems overall sound although I didn't carefully check all the derivations in the appendix. - I implemented the "OVIS_MC" gradient on a simple Gaussian model and in such a simple model, it works quite well, even with $S=1$ samples. - However, I did not manage to implement the "OVIS_~" gradients because they are not clearly enough stated: only the control-variate $c_k$ is stated in the manuscript but from the preceding text, it is not clear whether the terms $d_k$ need to be modified as well. Assuming that $d_k$ are as originally defined in Eq. 2, the "OVIS_~" gradients include terms of the form $\log(1-v_k)$ where $v_k$ is the self-normalised importance weight. If one of the self-normalised weights is numerically equal to $1$ the algorithm breaks down.

Clarity: - I think the paper is overall well written (given the space constraints). Yet, there are a few places which could be improved. In some places it is a bit difficult to keep track of all the approximations employed. - The paragraph "Optimal control for ESS limits and unified interpolation" would benefit from some more motivation. - The "OVIS_~" gradient should be stated explicitly (not just the expression for its control variate) for the reasons outlined above.

Relation to Prior Work: - The literature review seems satisfactory. However, in the numerical studies, I would like to know whether the "sleep-phase" update has been used within the RWS algorithm or not. If so, I would like to see results for the the RWS algorithm without this sleep-phase update given that there seem to be some previous works which suggest that leaving out this sleep-phase update step improves performance. - The manuscript should stress more clearly that for reparametrisable models, there exist alternative IWAE gradients which can perform very well but are not mentioned in the manuscript, e.g. the "STL" IWAE gradient from [R1] which was shown to work well in the simulations in [R2]. [R1] Roeder, G. et al. (2017). Sticking the landing: Simple, lower-variance gradient estimators for variational inference. NeurIPS. [R2] Tucker, G. et al. (2018). Doubly reparameterized gradient estimators for Monte Carlo objectives. arXiv:1810.04152.

Reproducibility: No

Additional Feedback: - I don't understand the use of "non-trivial" above Eq. 15 - Eq. 16: perhaps put parenthesis around the argumeng of $\log$ ### EDIT after rebuttal ### After reading the other reviews and the rebuttal, I am still positive about the main control-variate construction proposed in this paper. However I am going to lower my score to 5 because if weight clipping/truncation or scheduling/tempering is necessary to get a proposed importance-sampling type estimator to work in practice, this needs to be made much clearer in the paper, especially if code is not provided.


Review 2

Summary and Contributions: The paper presents a control variate method for score function gradient of the importance weighted lower bound (IWLB) and argues that the method can lead to a non-diminishing signal-to-noise ratio which is a main limitation of IWLB. The usefulness of the method is assessed through a range of numerical examples.

Strengths: This is a nice contribution as it's well-known that the score function gradient suffers from a large variance and that it's challenging to develop efficient control variates for IWLB. The contribution to making the SNR non-diminishing is also nice as it makes IWLB method more appealing. Its research is therefore relevant to the NeurIPS community.

Weaknesses: The proof and arguments are somewhat heuristic. The derivation in Section 4 involves many approximations so it'd be good if the authors provide some numerical examples examining the effect of their final control variate in Eq (12). That is, I would add an experimental study to compare (12) with the naive version without using control variate.

Correctness: The empirical study sounds reasonable and correct.

Clarity: Yes

Relation to Prior Work: Yes

Reproducibility: Yes

Additional Feedback: l79: Z hasn't been introduced


Review 3

Summary and Contributions: This paper proposes a new control variate for the score-function gradient estimator for the importance weighted autoencoder (IWAE) objective which has a signal-to-noise ratio that *increases* with the number of particles. This is unlike the state-of-the-art VIMCO gradient estimator in which case biased gradient estimators like reweighted wake-sleep (RWS) had to be used to make learning of the inference network effective. The proposed control variate fixes this issue and confirms this empirically.

Strengths: Making IWAE-like objectives amenable to good learning of the inference networks for discrete latent variables is an important problem because we can get both better learning of generative models and inference networks by increasing number of particles. So far, this has only been addressed using biased estimators like reweighted wake-sleep which instead alternates between optimizing two objectives. This paper theoretically grounds the new control variate by deriving a control variate that directly maximizes the signal-to-noise ratio. The resulting estimator is an approximation to this control variate which is empirically verified to be effective on a standard suite of experiments in previous papers ranging from gradient analysis on simple problems to deep generative models. In all cases, the new control variate convincingly improves previous approaches.

Weaknesses: none

Correctness: I haven't carefully went through the derivations but the motivation behind them and the final result make sense to me. My confidence stems from convincing empirical evaluation.

Clarity: Yes.

Relation to Prior Work: Yes. I particularly liked how they discussed what is the thing responsible for the improvement from VIMCO.

Reproducibility: Yes

Additional Feedback: None


Review 4

Summary and Contributions: The authors introduce a score function gradient estimator of the IWAE bound whose signal to noise ratio increases as K^{1/2} in the number of samples K. To this end the authors develop a control variate by optimizing the trace of the gradient's covariance matrix. To compute the control variate in practice they authors propose some simplifying approximations which put together result in the OVIS_MC gradient estimator. The authors further observe that additional approximations can be made in the regime of small (~=1) and large (>>1) effective sample sample sizes respectively and propose OVIS~ a unifying gradient estimator. The empirical evaluation shows that OVIS is 1) able to achieve the theoretical SNR when training a Gaussian latent variable model, 2) is able to train inference networks for a GMM which are close (L2 distance) to the true posterior, and 3) performs competitively with other reparameterization-free gradient estimator for training DGMs.

Strengths: Score-function gradient estimators are important for a variety of discrete latent variable models, e.g. in reinforcement learning or probabilistic programming. Thus improving the training and quality of these models is of interest to the broader NeurIPS community. To the best of my knowledge all claims made in the paper are sound and the analysis and empirical evaluation are sufficient to demonstrate improvements over state-of-the art reparameterization-free gradient estimators.

Weaknesses: While it is reassuring to the that OVIS variants are able to achieve the theoretical SNR in experiment 1 it is not surprising that they outperform the VIMCO baseline and pathwise-IWAE estimator in terms. Additional comparing to the DReG-IWAE gradient estimator [1] would draw a more complete picture. In experiment 3 it would also be interesting to see how OVIS performs compared to reparameterized and doubly reparameterized estimators (DReG-IWAE and RWS-IWAE) [1] as well. [1] George Tucker, Dieterich Lawson, Shixiang Gu, and Chris J Maddison. Doubly reparameterized gradient estimators for monte carlo objectives. arXiv preprint arXiv:1810.04152, 2018.

Correctness: To the best of my knowledge the claims and empirical methodology are correct.

Clarity: The paper is well structured and clearly written. The authors do a good job in guiding the reader through this fairly technical paper.

Relation to Prior Work: The authors appropriately discuss prior work and explain how it differs from their contribution.

Reproducibility: Yes

Additional Feedback: There is a typo on the end of line 133. I assume $w_k >> ...$ should be $w_{k'} >> ...$. **Update** I've carefully read through the author's rebuttal and the other reviews. My score remains unchanged.

[Author Response · NeurIPS 2020]

We thank all reviewers for their constructive comments. Clarity and presentation inputs will be used in the final version.
All reviewers confident [scores 4,4,4,3] and concur that OVIS is a novel control variate for the score function estimator
for importance weighted bounds and that, unlike existing IW score-based estimators (VIMCO, Reinforce) and unlike
the basic pathwise IWAE, it allows for more efficient inference network learning because its SNR increases with K.
Reviewers found that OVIS improves the learning of reparameterization-free DGMs [R1,R3,R4] and the experiments
were sufficient to convince the reviewers [R2, R3, R4]. Except for the section on $\mathrm{OVIS}_\sim$ [R1], all the reviewers have
found the paper to be well-written and clear. We address your concerns below.

**Reweighted Wake Sleep [R1]**   We use the RWS algorithm with wake-$\phi$ updates, which have been found to be more
effective than sleep-$\phi$ updates.

**Biased induced by the use of the IWR bound [R1]**   The scheduling scheme for the experiment 6.3 was indeed
crucial for the Sigmoid Belief Network, but not for Gaussian VAE, which performed well using a fixed value $\alpha = 0.7$.

**Ablation Study [R2]**   Studying Reinforce and VIMCO acts as an ablation study for OVIS.

**Implementation for $\mathrm{OVIS}_\sim$ [R1]**   Numerical instabilities can be avoided by clipping the normalized weights. The
full code will be released upon publication. Eq. (17) relies on Eq. (3). This will be explicitly stated in the main text.

**Complexity of $\mathrm{OVIS}_{\mathrm{MC}}$ and motivations for $\mathrm{OVIS}_\sim$ [R1]**   $\mathrm{OVIS}_{\mathrm{MC}}$ has complexity $\mathcal{O}(K + S)$ because the $S$
samples for estimating the $K$ baselines can be reused. $\mathrm{OVIS}_\sim$ has a complexity $\mathcal{O}(K)$. Estimating $\phi$ using $\mathrm{OVIS}_{\mathrm{MC}}$
requires a budget of $K' = K + S$ particles. The ratio $S/K$ is a trade-off between the tightness of the bound $\mathcal{L}_K$ and the
variance of the control variate estimate. The manuscript focuses on studying the sole effect of the control variate given
the bound $\mathcal{L}_K$. For the other estimators, this is a sub-optimal use of the budget $K'$ because $\mathcal{L}_{K'} \geq \mathcal{L}_K$. $\mathrm{OVIS}_\sim$ both
validates our asymptotic results and bypasses the burden of requiring auxiliary samples. By contrast with the previous
experiments, we trained the Gaussian VAE using the budget $K'$ optimally (i.e. relying on $\mathcal{L}_{K'}$ whenever no auxiliary
samples are used). We observed that $\mathrm{OVIS}_\sim$ outperforms $\mathrm{OVIS}_{\mathrm{MC}}$ despite the generative model is evaluated using
$\mathcal{L}_{K'}$ in all cases (figure 1). This experiment will be detailed in the Appendix.

Figure 1: Training the Gaussian VAE model with a fixed and optimally used particle budget $K' = K + S$ and $\alpha = 0.7$.

**Extended comparison with reparameterization-based IWAE estimators [R1, R4]**   Although we focus on
reparameterization-free methods, we concur that advanced reparameterization-based IWAE estimators (STL and
DReG) should be added to the experiments. Comparing OVIS with the pathwise estimator also requires studying the
use of the IWVR bound ($\alpha > 0$) for both estimators. We updated the experiment and 6.3 accordingly (figure 2).

Figure 2: Training the Gaussian VAE (3 seeds) using $\mathrm{OVIS}_\sim$ with $\alpha \in \{0, 0.7\}$ and using multiple baseline estimators.

[Meta-Review · NeurIPS 2020]

This paper is accepted, however, there are a few issues that require revision. All reviewers agreed that OVIS is a novel control variate that resolves an important open problem. However, the use of the IWR bound muddled the experimental conclusions and lead to confusion. Carefully separating out the effects of OVIS and using the IWR bound is critical for improving this paper. R1 was concerned about the sensitivity to the annealing scheme and the annealing details. Furthermore, this made it challenging to compare methods in some experiments. Figure 2 in the rebuttal is a step in the right direction, and based on that effort, I believe that the authors can clarify the main paper. R1 asked about numerical instabilities and the authors explained that they clipped the importance weights. This introduces bias and potentially complicates experimental conclusions. Given that the expression in question appears in VIMCO and can be computed numerically stably without clipping, I ask that the authors rerun their experiments without clipping or provide evidence that the bias from clipping is limited. R1 asks about the computational complexity of the method. While, we agree that the method requires O(K+S) samples and expensive function evaluations, evaluating the estimator potentially requires O(KS) operations. The authors should explain how to compute this in O(K+S) operations or clarify this. While the reviewers found the paper clear, I did not find the notation clear. The authors should carefully balance defining new variables versus writing them out.